A comparative study on the lipid layer thickness analysis of medical staff before and after work

Li Qian
Liu Xiufen
Ren Yu
He Tianlong
Shao Fei
Yimingtuohuti Nuerailimu
Li Dan
Lu Chengwei lcwchina800@jlu.edu.cn
Department of Ophthalmology, The First Hospital of Jilin University , Changchun, Jilin , China
Redondo Beatriz
Electronic publication date: 2024 Oct 31
Publication date: 2024
Volume: 12
Electronic Location ID: e18258
Received 2024 Apr 24; Accepted 2024 Sep 17
Copyright year: 2024
License: This is an open access article, free of all copyright, made available under the Creative Commons Public Domain Dedication. This work may be freely reproduced, distributed, transmitted, modified, built upon, or otherwise used by anyone for any lawful purpose.
License URL: https://creativecommons.org/publicdomain/zero/1.0/

Keywords: Lipid layer thickness, Dry eye disease, Medical staff, Video display terminal, Gaush iDea Ocular surface analyzer

Funding: National Natural Science Foundation of China 82171023 Science and Technology Department of Jilin Province 20210101279JC Bethune Center for Medical Engineering and Instrumentation BQEGCZX20210XX Tianhua Health Public Welfare Foundation of Jilin Province, China J2022JKJ027 This research was supported by the National Natural Science Foundation of China (No. 82171023), the Science and Technology Department of Jilin Province Fund (No. 20210101279JC), the Bethune Center for Medical Engineering and Instrumentation (No. BQEGCZX20210XX), and the Tianhua Health Public Welfare Foundation of Jilin province, China (No. J2022JKJ027). The funders had no role in study design, data collection and analysis, decision to publish, or preparation of the manuscript.

==============================
Background & Aims

To study the change of the lipid layer thickness analysis in medical staff (MS) before and after work, and to explore the significance of measuring lipid layer thickness (LLT) respectively in four quadrants.

Methods

Ocular Surface Disease Index (OSDI) questionnaire and video display terminal using time for 55 MS were collected (the informed consent was obtained from all patients). Noninvasive tear break-up time, LLT, tear meniscus height (TMH), meibomian glands (MG) dropout, and blink pattern before and after work (worked for more than 4 h) of 110 eyes were measured by Gaush iDea Ocular surface analyzer. Lid margin abnormalities were evaluated by the slit-lamp microscopy.

Results

The average OSDI score of 55 MS was 25.68 ± 14.91. The average LLT of 110 eyes after work (65.12 ± 3.63 nm) was significantly reduced compared to before work (66.54 ± 4.16 nm), p < 0.05. The LLT in the superior quadrant was significantly thinner than that in the other three quadrants, p < 0.01. The average LLT was positively correlated with the LLT in the inferior (r = 0.822, p < 0.001), nasal (r = 0.261, p < 0.001), and temporal quadrant (r = 0.372, p < 0.001), while was negatively correlated with the MG dropout in lower lid (r = −0.209, p = 0.002). There was a significant correlation between the LLT in the inferior quadrant and the VDTt (r = −0.173, p = 0.01). The LLT of inferior quadrant were positively related to the TMH (r = 0.149, p = 0.027) and negatively related to MG dropout in lower lid (r = −0.162, p = 0.017).

Conclusion

The LLT significantly decreases after work in MS. The distribution of the lipid layer on the ocular surface is uneven. It is unreasonable for current detection instruments to measure the inferior quadrant LLT alone to represent average LLT.

Introduction

Dry eye disease (DED) is a disease of the tear and ocular surfaces causing symptoms of discomfort, visual disturbance, and tear film instability (Craig et al., 2017). The prevalence estimates are ranging from 5% to 50% across populations. The incidence of DED may vary between different professions, different regions of the world, different living standards. Nowadays, dry eye has become one of the most important eye diseases that troubles the working population. The latest research results illustrated that the risk factors of DED include allergic conjunctivitis, age, air pollutions, meteorological conditions in cold regions, etc., (Lu et al., 2021, 2023). Besides, the use of video display terminal (VDT) was considered as an important risk factor for DED. With the comprehensive application of electronic medical records, medical staff (MS) need to face electronic terminals for a large amount of work time every day. Besides the prolonged use of VDT, frequent night shifts and highly intensive use of surgical microscopes in MS, and the overwhelming working pressure may aggravate the instability of the tear film, leading to the increase of the DED. In addition, the working environment of MS is relatively dry and closed, and all of which may be factors that can lead to DED (Craig et al., 2017; Lu et al., 2021). In Chan, Chuang & Wong’s (2021) study, 68.2% healthcare workers had subjective symptoms of DED, which is higher than other investigation. When MS suffer from dry eyes, they may blink frequently and shed tears due to ocular discomfort, resulting in a decrease in work efficiency. Especially, symptoms of visual fluctuations may increase medical risks.

The normal function of ocular surface is maintained by homeostasis of the stable and sufficient tear film. Based on the definition of Tear Film and Ocular Surface Society Dry Eye Workshop (TFOS DEWS) II (Wolffsohn et al., 2017), the etiological of DED contained tear film instability and hyperosmolarity, ocular surface damage and inflammation, and neuro-sensory abnormalities, with the tear film instability being the most important one. The outermost layer of the tear film is the lipid layer, with the thickness of approximately 40–100 μM, which maintains the stability of the tear film and reduces tear evaporation (Bai, Ngo & Nichols, 2019). Blinking applies a shearing force that lowers the viscosity, making the meibomian lipids easier to eject from the meibomian orifices. The meibum secreted by meibomian glands (MG) is the main source of lipids in the tear film lipid layer (TFLL) (Knop et al., 2011). In the meibomian gland dysfunction (MGD), the keratinize of the MG orifices and loss of meibocyte progenitor cells lead to an alteration of the quality and quantity of the secreted lipids (Jester, Parfitt & Brown, 2015). All these changes trigger the abnormal tear evaporation, decreased tear film stability, and result in the occurrence of DED (Khanal et al., 2021).

Gaush iDea Ocular surface analyzer (SBM Sistemi S.r.l. Inc, Torino, Italy) and LipiView interferometer (TearScience Inc, Morrisville, NC, USA) are both dry eye related examination device which can provide quantitative values of the lipid layer thickness (LLT). Gaush iDea is a device which can provide comprehensive examinations for the diagnosis of DED, including non-invasive tear break-up time (NIBUT), LLT, tear meniscus height (TMH) and blink pattern. So far, this is the only machine on the market that can divide the lipid layer into different four quadrants (superior, nasal, inferior, temporal) and measure them separately. It can also provide infrared images for evaluating MG dropout and catheter dilation (Zang et al., 2018). However, LipiView can only provide the results of the lower quadrant LLT and blink analysis, which has a significant functional deficiency. Thus, Gaush iDea was employed in this study.

Gaush iDea consists of a main body, which contains the camera and the circuit with LEDs for lighting, and two different magnetic cones to be applied on the front of the device depending on the examination desired. It is also equipped with a mechanical yellow filter, activated by a lever on the side of the device, which allows a better view of the blue light projected by the device on the eye of the patient. All the examinations are performed thanks to the light emitted by the device, passing to a cone, reflecting on the eye and captured by the camera. Depending on the cone chosen the light will be reflected with different behaviors by the tear film. The software can work out the thickness of the lipid layer based on the colors observed according to the study of Dr. Guillon (Remeseiro et al., 2014).

Therefore, in order to study the change of the ocular surface status in MS before and after work, Gaush iDea was used as the research tool to analyze the NIBUT, LLT, TMH, MG dropout and blink pattern. And what’s more, for the first time in the research, the lipid layer was divided into four quadrants (superior, nasal, inferior, temporal), in order to deeply explore the relationship between the lipid layer and tear film stability.

Materials and methods

Participants are MS working in the first hospital of Jilin University, China. The occupation are physicians or nurses, age from 23 to 59. 110 eyes of 55 participants were enrolled into this study. The sample size was calculated with PASS.15 software, based on a LLT difference between groups of 1.4 ± 4 SD, to achieve an effect size (d) of 0.35, a significance level (α error prob) of 0.05, and a power (1−β err prob) of 0.8, in a two-tailed test, yielding a sample size of 67 participants in each group. They voluntarily participated in this study and the recruitment is random. Autoimmune systemic diseases, ocular diseases except cataract, and past history of ocular surgery were excluded. This study procedures were performed in accordance with the principles of the World Medical Association Declaration of Helsinki. Ethical approval was obtained from the Ethics Committee at the First Hospital of Jilin University (ethical approval number: 22K017-001). The written informed consent was obtained from all patients.

All participants registered basic information such as age, gender, and date of birth and completed Ocular Surface Disease Index (OSDI) questionnaire. We used the OSDI formula to calculate and determine the severity of DED, where OSDI = the total score of all questions × 25/the total number of questions answered. Value less than 13 was considered normal, value 13–22 was considered mild, value 23–32 was considered moderate, and value greater than 33 was considered severe DED. We investigated the average daily VDT using time (VDTt) of participants over the past 3 months. In the questionnaire, it contains how many hours did the participate use mobile phones, televisions, various types of computers and other types of VDT every day and we added the hours together.

The same doctor measured the TMH, NIBUT, MG dropout, blink pattern and LLT before and after work (work for more than 4 h) respectively on the same day by Gaush iDea Dry eye diagnostic device. The device uses the Dr. Guillon classification (Zang et al., 2018) in a dynamic approach, the software measure and grade all frames of the blinking interval from once the eye is almost totally open to totally closed mapping the average thickness of all the frame and creating an average result between all of them. LLT was measured in different directions by rotating the magnetic cone. The algorithm uses multiple algorithms to define area of interest, color adjustment and fringe type. The lipid layer is an oily compound that deflects light based on its thickness. The camera acquires the deflects light, and the software can work out the thickness of the lipid layer based on the colors (Fig. 1) observed according to the study of Dr. Guillon. The TMH measurement results were the average of more than three times at different positions of the tear river(Fig. 2).

Figure 1 Illustration of LLT in different quadrants by Gaush iDea dry eye diagnostic device.

(A) LLT in superior quadrant; (B) LLT in inferior quadrant; (C) LLT in nasal quadrant; (D) LLT in temporal quadrant.

Figure 2 Representative illustration of MG photography and MG dropout score.

Lid margin abnormalities were detected by the slit-lamp microscopy. The results were scored from 0 through four according to the findings present. We observed irregular lid margin, vascular engorgement, plugged MG orifices, and anterior or posterior replacement of the mucocutaneous junction.

Statistical analyses were performed using the SPSS® for Windows software (version 22.0; SPSS Inc., Armonk, NY, USA). Data for all parameters were presented as mean ± standard deviation. According to the distribution characteristics of numerical values, statistical methods of t-tests, analysis of variance, and rank sum tests were used. Spearman’s correlation coefficient (r) was used to evaluate the correlations among different parameters. A p-value less than 0.05 was considered statistically significant.

Results

Characteristics of participants

A total of 55 MS (male 10, female 45) of 110 eyes were enrolled in this study. Mean age was 35.72 ± 10.10. The average VDTt was 9.00 ± 3.90 h per day. The average OSDI score was 25.68 ± 14.91. The average NIBUT was 8.13 ± 1.54 s. The average TMH was 0.20 ± 0.10 mm. The average MG dropout in upper eyelid was 21.03 ± 12.62%, and in lower eyelid was 36.59 ± 15.67%. According to the OSDI score, there were eight individuals without DED, 15 individuals (27.3%) with mild DED, 18 individuals (32.7%) with moderate DED, and 14 individuals (25.5%) with severe DED. The overall prevalence rate is 85.5%. The OSDI score of the group of age ≥41 (33.31 ± 17.46, n = 13) is significantly greater than the group of age between 21–30 (20.70 ± 11.76, n = 19), p = 0.016, the difference is statistically significant. However, there was no significant difference in OSDI scores between participants in the 31–40 age group (27.20 ± 13.64, n = 23) and the group of age ≥41 or the group of age between 21–30, p > 0.05.

Comparison of ocular surface characteristics before and after work

Ocular surface characteristics were measured before and after work. The average time interval between two measurements are 4.81 ± 0.83 h, during which the participants engaged in medical activities such as treating patients, communicating, and writing medical records. The results (Table 1) showed that the average LLT of the right and left eyes after work were significantly reduced compared to before work, p < 0.05, the difference was statistically significant. Furthermore, we combined the results of the average LLT of both eyes and compared and analyzed the changes in average LLT before and after work (n = 110). The average LLT was 66.54 ± 4.16 nm before work, and 65.12 ± 3.63 nm after work. The results showed that the average LLT of both eyes after work was significantly reduced compared to before work, p < 0.05, with a statistically significant difference. There are no differences in TMH, NIBUT and partial blink rate (PBR) before and after work (p > 0.05).

Table 1 Comparison of eye surface analysis results before and after work.

	Before work	After work	p value	
LLT of both eyes (nm)	66.54 ± 4.16 (n = 110)	65.12 ± 3.63 (n = 110)	0.001	
LLT of right eyes (nm)	66.84 ± 4.34 (n = 55)	65.53 ± 3.90 (n = 55)	0.038	
LLT of left eyes (nm)	66.24 ± 3.99 (n = 55)	64.71 ± 3.32 (n = 55)	0.011	
TMH (mm)	0.20 ± 0.12 (n = 110)	0.20 ± 0.08 (n = 110)	0.430	
NIBUT (s)	8.16 ± 1.52 (n = 110)	8.10 ± 1.56 (n = 110)	0.748	
PBR (%)	12.25 ± 12.04 (n = 110)	13.51 ± 12.19 (n = 110)	0.370	
Notes:

LLT, lipid layer thickness; TMH, tear meniscus height; NIBUT, noninvasive tear break-up time; PBR, partial blink rate.

Comparative analysis of the mean LLT in each quadrant

To validate the distribution pattern of the tear film lipid layer, we conducted an analysis of the lipid layer thickness (LLT) across the superior, inferior, nasal, and temporal quadrants of both eyes (n = 110), as presented in Table 2. Utilizing the Wilcoxon signed-rank test, we compared the variations among these four quadrants. Our findings revealed that the LLT in the superior quadrant was significantly thinner than that in the inferior quadrant (c = −2.623, p = 0.009), the nasal quadrant (c = −4.586, p < 0.001), and the temporal quadrant (c = −5.618, p < 0.001), with statistical significance evident in each comparison. However, no statistically significant difference was observed between the nasal and temporal quadrants (c = −0.796, p = 0.426). Similarly, no significant difference was detected between the inferior quadrant and either the nasal (c = −0.668, p = 0.504) or temporal quadrants (c = −0.971, p = 0.332).

Table 2 LLT in four quadrants.

LLT in four quadrants (n = 110)	Median (P25, P75)	
LLT of superior quadrant	65.5 (64.3, 66.5)	
LLT of inferior quadrant	66.3 (63.0, 75.2)	
LLT of nasal quadrant	66.3 (65.0, 68.0)	
LLT of temporal quadrant	66.4 (65.3, 67.5)	

Correlation analysis of dry eye parameters

We applied Spearman correlation analysis to test the correlation of DED parameters (Table 3). The average LLT was positively correlated with the LLT in the inferior (r = 0.822, p < 0.001), nasal (r = 0.261, p < 0.001), and temporal quadrant (r = 0.372, p < 0.001), but not superior quadrant (r = 0.014, p = 0.839). The LLT was negatively correlated with the MG dropout in lower lid (r = −0.209, p = 0.002) and there was a significant correlation between the LLT in the inferior quadrant and the VDTt (r = −0.173, p = 0.01). The LLT in the inferior quadrant was positively related to the TMH (r = 0.149, p = 0.027) and negatively related to MG dropout in lower lid (r = −0.162, p = 0.017), while the thickness of the other quadrants was not related to the above parameters. The rate of MG dropout was negatively correlated with the NIBUT (r = −0.225, p = 0.001 in upper lid and r = −0.165, p = 0.014 in lower lid), but not with the TMH. The results showed that age was significantly correlated with VDTt (r = −0.214, p = 0.001), OSDI score (r = 0.316, p = 0.001), LLT (r = 0.186, p = 0.06), TMH (r = 0.155, p = 0.022), PBR (r = 0.243, p = 0.001), and MG dropout (r = 0.279, p = 0.039).

Table 3 Correlation analysis of dry eye related examination parameters.

		LLT	SLLT	ILLT	NLLT	TLLT	TMH	NIBUT	PBR	MG(UL)	MG(LL)	LMA(UL)	LMA(LL)	OSDi	VDTt	
Age	r	0.186**	0.062	0.171*	0.079	0.081	0.155*	−0.067	0.216**	0.243**	0.109	−0.048	−0.016	0.316**	−0.214**	
	p	0.006	0.358	0.011	0.245	0.23	0.022	0.325	0.001	<0.001	0.106	0.476	0.812	<0.001	0.001	
LLT	r		0.014	0.822**	0.261**	0.372**	0.091	0.028	0.074	−0.006	−0.209**	0.066	0.011	0.043	−0.076	
	p		0.839	<0.001	<0.001	<0.001	0.181	0.685	0.273	0.924	0.002	0.33	0.873	0.525	0.26	
SLLT	r			−0.075	0.022	0.045	−0.045	−0.001	−0.037	−0.006	−0.051	−0.016	−0.074	0.1	0.13	
	p			0.27	0.74	0.509	0.51	0.994	0.581	0.93	0.456	0.819	0.273	0.138	0.055	
ILLT	r				0.131	0.259**	0.149*	0.047	0.091	−0.052	−0.162*	0.08	0.062	−0.013	−0.173*	
	p				0.052	<0.001	0.027	0.484	0.177	0.442	0.017	0.237	0.362	0.853	0.01	
NLLT	r					0.141*	−0.078	0.003	−0.043	−0.102	−0.05	−0.04	−0.07	0.146*	0.018	
	p					0.037	0.252	0.97	0.522	0.13	0.464	0.559	0.299	0.03	0.795	
TLLT	r						0.016	−0.008	0.09	0.042	0.007	0.017	−0.048	0.079	−0.035	
	p						0.815	0.909	0.182	0.533	0.917	0.807	0.475	0.241	0.601	
TMH	r							0.142*	0.031	0.11	−0.036	0.009	−0.124	0.049	−0.092	
	p							0.035	0.644	0.104	0.6	0.898	0.066	0.467	0.175	
NIBUT	r								−0.01	−0.225**	−0.165*	−0.123	−0.092	0.045	−0.076	
	p								0.888	0.001	0.014	0.068	0.172	0.504	0.264	
PBR	r									−0.106	−0.034	0.114	−0.1	−0.007	0.081	
	p									0.117	0.611	0.092	0.14	0.924	0.231	
MG (UL)	r										0.195**	−0.144*	−0.149*	−0.02	0.02	
	p										0.004	0.032	0.027	0.766	0.763	
MG (LL)	r											0.054	0.228**	0.004	0.194**	
	p											0.429	0.001	0.956	0.004	
LMA (UL)	r												0.330**	−0.072	−0.06	
	p												<0.001	0.29	0.374	
LMA (LL)	r													0.148*	−0.012	
	p													0.029	0.862	
OSDI	r														−0.089	
	p														0.188	
Notes:

LLT, lipid layer thickness; SLLT, LLT of superior quadrant; ILLT, LLT of inferior quadrant; NLLT, LLT of nasal quadrant; TLLT, LLT of temporal quadrant; TMH, tear meniscus height; NIBUT, noninvasive tear break-up time; PBR: partial blink rate; MG (UL), meibomian glands dropout in upper lids; MG (LL), meibomian glands dropout in lower lids; LMA (UL), lid margin abnormalities in upper lids; LMA (LL), lid margin abnormalities in lower lids; OSDI, ocular surface disease index; VDTt, video display terminal usage time. *: p < 0.05, **: p < 0.01.

Discussion

DED is the most common ocular disease in nowadays. Based on the definition of Tear Film and Ocular Surface Society Dry Eye Workshop (TFOS DEWS) II (Wolffsohn et al., 2017), the tear film instability plays a key role in the pathogenesis of DED. The lipid layer is the external layer of the tear film, and it is an oily secretion which is formed in the meibomian glands (MG). Analyzing the interference patterns of the tear film lipid layer can help the experts to diagnose dry eye. Previous studies have reported that a thin tear film lipid layer is related to severe dry eye symptoms and can disturb the stability of tear film (Craig & Tomlinson, 1997). Less or equal to 75 nm of the LLT can diagnose MGD and has high sensitivity and specificity (Blackie et al., 2009). But Cohen et al. (2020) using a Tear Film Imager to measure the distribution of the mucoaqueous and lipid layers of the tear film, they got the conclusion that the lipid layer appears nonuniform in the lipid-deficient dry eye, whereas in the normal control group is uniform. Bai et al. (2022) also verified the fact that the lipid layer is unevenly distributed in the tear film and concluded that the measurement of the average LLT cannot fully represent the condition of the tear film.

To the best of our knowledge, this is the first study that the LLT was analyzed in different four quadrants (superior, nasal, inferior, temporal) separately and the result revealed that the distribution of tear lipid layer is uneven. The LLT in superior quadrant was significantly thinner than that of the LLT in the other three quadrant, while this difference does not exist in the comparison between the inferior, nasal and the temporal. We presumed this may be related to the gravitational effect of tears. Under the force of gravity, tears are more distributed below the ocular surface than above it, leading to drainage into the nasal cavity from the lacrimal dots. And Hyeonha Hwang et al. (2017) developed Lipiscanner 1.0 system for quantitative measurements of the LLT. They found that the closer to the lower eyelid, the thicker the LLT. But their results were also limited to only detecting the results in the inferior quadrant of the eye surface, without the LLT in the superior, temporal and nasal quadrant. Moreover, our correlation analysis results also indicate that the LLT in the inferior quadrant is correlated with the TMH, indicating that when the amount of tear increases, the LLT in inferior quadrant increases. Therefore, for cases of increased reflective tear secretion, the results of the LLT in the inferior quadrant cannot fully represent the lipid layer state on the ocular surface. Therefore, it is necessary to refer to the examination results of the other quadrants.

According to the conclusion of Lee, Hyon & Jeon (2021) the average LLT should not be regarded as a stable physiologic condition. Compared to the thin- and normal-LLT groups, the eyes in the thick-LLT group (LTT = 100 nm) had a significantly shorter TBUT, higher ocular staining score, and higher OSDI. It is believed that reflex tear secretion may lead to higher LLT. The result of the average LLT with corneal erosions was thicker than those without erosions, suggesting that the LLT cannot be used alone as a severity indicator for evaluating DED (Lee, Hyon & Jeon, 2021). Therefore, our results show that LLT below is associated with tear secretion, which can explain the reasons for the above situation. Moreover, King-Smith et al. (2013) found that the rupture of the tear film appeared around the edges of thick but not thin TFLL areas. Therefore, for the complex relationship between the LLT and the overall tear film health, we need to comprehensively evaluate various ocular surface indicators, rather than relying solely on a single indicator in the diagnosis of DED.

Prolonged VDTt was considered as an important risk factor for DED, and it can reduce blink rates and increase incomplete blinks, leading to tear film instability, and resulting in increased incidence of DED (Kamoy et al., 2022). However, to date, the research regarding to the correlation between LLT and the VDTt was rarely reported. Our research indicated a negative correlation between LLT in the lower quadrant and the VDTt, with participants who used VDT for a longer time per day having thinner LLT in the lower quadrant. Besides VDTt, long term exposure to unventilated and lower relative humility environments is also considered as an important factor for DED (Abusharha & Pearce, 2013), which the MS face every day. Moreover, MS who regularly worked night shifts had more severe and frequent dry eye symptoms than those daytime workers (Zhang et al., 2020). In our survey, dry eyes were common. Therefore, we believe that after prolonged medical work, the condition of the eye surface will become worse.

The prevalence of DED may vary due to different regions of the world, with the prevalence estimates ranging from 5% to 50% across populations (Lu et al., 2023). We determined the condition of dry eye among MS through the OSDI questionnaire, and determined the relevant indicators of dry eyes through examination of NIBUT, TMH, and LLT. In our results, the proportion of DED among MS was indeed significantly higher than other investigations. The results of NIBUT, TMH, and LLT before and after work are evaluated. The results showed that LLT became thinner after work for 4 h. This also reflects that LLT is not a stable indicator. For the lipid layer was important to maintain the tear film stability, this result suggests that fatigue eye use may lead to a decrease in tear film stability. We can use the conclusion as a basis to guide DED treatment. But there was no significant change in NIBUT and TMH and blink pattern. We analyzed that the result may be an increase in reflective tear secretion due to the thinning of LLT, which has been previously proposed by researchers. The change in NIBUT may be a slow process that requires long-term observation.

The tear film is primarily derived from the MG. The lipid layer is compressed towards the lower lid during a blink, then spreads upward as the lid opens. The shape of MG is a sensitive index for the diagnosis of early MGD (Adil et al., 2019). According to our data, there was a significant correlation between LLT and MG dropout. This manifestation was more evident in the correlation analysis between LLT in the lower quadrant and lower eyelid MG dropout. Therefore, our analysis shows that the loss of lower eyelid MG has a greater impact on LLT. Correlation analysis also showed that LLT in the inferior quadrant was related to the TMH and the MG dropout, while LLT in other quadrants were not related to the above parameters. The average LLT was significantly positively correlated with LLT of inferior, nasal temporal quadrants, but not with LLT of superior quadrant. Therefore, we believe that LLT in each quadrant should be evaluated separately. In our opinion, under normal circumstances, the average LLT measurement should include the lower, nasal, and temporal quadrants. When reflective tear secretion increases such as keratitis, the upper LLT should be also taken into account. However, the detection algorithm of the instrument we use calculates the average of the four quadrant LLT. While the commonly used LipiView II measurement range is 5.0 mm × 2.5 mm in the inferior quadrant, which cannot accurately reflect LLT. Therefore, we believe that the current research instrument should be improved in algorithm.

Besides, in our study, there were a significant correlation between age and OSDI score, LLT, NIBUT, TMH, MG dropout, and lid margin abnormalities. This was consistent with the previous research results (Den et al., 2006). With age increasing, the MG shrinks and the anatomical structure of the lid margin changes, leading to a decrease in lipid secretion and an increase in the incidence rate of DED. Wu & Chang’s (2022) study suggests that DED may be related to dermatochalasis, for elderly females with higher LLT were less severe dermatochalasis. Therefore, age induced MG atrophy and skin problems are the causes of DED that should be considered.

But there are still some limitations in our study. Firstly, the number of volunteers in this study is relatively small. Moreover, the uneven age distribution and differences in working conditions may be another drawback of this study. Another limitation is that the OSDI questionnaire assesses individuals in binocular conditions (55 subjects). However, for the other variables, data from both eyes (110 eyes) were included, even though the two eyes of the same patient may be correlated. But the variance between eyes is generally smaller than the variance between different individuals. Consequently, combining measurements from both eyes can lead to an underestimation of the true variance, increasing the risk of a Type 1 error.

Conclusion

In conclusion, the LLT significantly decreases after work in MS, and the distribution of the lipid layer on the ocular surface is uneven. The thickness of the superior lipid layer is significantly thinner than that of the inferior lipid layer. The average LLT measurement should include the lower, nasal, and temporal quadrants for the diagnosis of DED. But in special cases such as increased tear reflex secretion, attention should be paid to the results of LLT in other quadrants.

Supplemental Information

Supplemental Information 1 Raw data.

Additional Information and Declarations

Competing Interests

Author Contributions

Human Ethics

Data Availability

The authors declare that they have no competing interests.

Qian Li conceived and designed the experiments, analyzed the data, prepared figures and/or tables, authored or reviewed drafts of the article, and approved the final draft.

Xiufen Liu analyzed the data, authored or reviewed drafts of the article, and approved the final draft.

Yu Ren analyzed the data, prepared figures and/or tables, and approved the final draft.

Tianlong He performed the experiments, prepared figures and/or tables, and approved the final draft.

Fei Shao performed the experiments, prepared figures and/or tables, and approved the final draft.

Nuerailimu Yimingtuohuti performed the experiments, analyzed the data, prepared figures and/or tables, and approved the final draft.

Dan Li performed the experiments, authored or reviewed drafts of the article, and approved the final draft.

Chengwei Lu conceived and designed the experiments, authored or reviewed drafts of the article, and approved the final draft.

The following information was supplied relating to ethical approvals (i.e., approving body and any reference numbers):

This study was approved by the Ethics Committee of The First Hospital of Jilin University (Approval number: 22K017-001). The informed consent was obtained from all patients.

The following information was supplied regarding data availability:

The raw measurements are available in the Supplemental File.

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
