# Peer review of "A comparative study on the lipid layer thickness analysis of medical staff before and after work"

_PeerJ, doi:10.7717/peerj.18258_

## Round 0.1 · original submission · Major Revisions

· Academic Editor

Major Revisions

It has been reviewed by two experts in the field. Revisions are necessary before the manuscript is suitable for publication.

·

Basic reporting

NO COMMENT

Experimental design

In line 112, how did you investigate the data on the time of use of the VDTs, what questions did the questionnaire apply and did it include the use of the devices only at work or also at home? Did you evaluate time, type of device? Be more explicit in this part.

In line 143 about the difference found between the age groups is not clear, improve the wording of this paragraph

Validity of the findings

In the quadrant comparison analysis at line 156, I wonder what the intention was in analyzing the superior quadrants with the inferior quadrants and the nasal quadrants with the temporal quadrants; In my opinion, the results they obtained are as expected, because the tear lake rests in the lower quadrant and due to gravity there must be more tear volume and it must be thicker. In my opinion the results would be more relevant if the upper quadrants were compared. or inferior with the nasal and temporal, and it would be very interesting because these will be in the area of exposure to the opening of the eyelid.
In addition, there should be changes in the correlation analysis of the parameters.

·

Basic reporting

The English writing is generally fine, but I recommend giving a grammar check
8. Methods: Methods, in general it is written in the past, not in the present.
13. Discussion: line 181 “To analyzing the interference patterns of the tear film lipid layer can help the experts to diagnose dry eye”. Review general grammar in the manuscript. for example this statement. where should it be like this... Analyzing the interference patterns of the tear film lipid layer can help the experts to diagnose dry eye.
references: ok
manuscript structure: ok
figures and tables ok. but table 1 : Table one has very little information, and it is already mentioned in the text, consider eliminating it, and leaving only the text, or consider merging it with another table or including more information to make it more complete.
The title and abstract They are not good and I recommend changing the title and rewriting and improving the abstract as I will discuss later.
1. Title: The title of the manuscript does not represent the work performed. LLT measurement alone does not represent the status of the ocular surface. I would recommend changing the title to replace evaluation of the status of the ocular surface with LLT analysis instead

Experimental design

experiment is original,
The manuscript falls within the scope of the journal.
The title and abstract They are not good and I recommend changing the title and rewriting and improving the abstract as I will discuss later.
The results are clear and well-described.
ethics and methods are well described.
methods: Explain who performed the dry eye tests. Was it done only once or was it done several times and then an average was made or how it was done.
8. Methods: Methods, in general it is written in the past, not in the present.
9. Methods: write the sample size calculation for the objective of the study

Validity of the findings

The manuscript is novel, interesting, original.
The information and data are shown and it is explained how the statistical analysis was carried out.

Additional comments

2. Abstract: “iDea” veryfy this is the accurrate name…. or might me iDra?
3. Abstract: mention when it is evaluated and how many hours elapsed from the beginning to the end of working hours
4. Abstract abreviature (AVE) for LLT Average is not needed and is confusing, I recommend to only use the abrebviation LLT instead and to review the sentence and check that the statements are clearly understood because it is confusing.
5. Abstracrt, resoults: I recommend to include the values and umbers of the outcomes and the significance in p value as well as the values significance and strength of the correlation.
6. Abstract Include number of participants included
7. Abstract: abstract: in the manuscript many more evaluations are mentioned such as osdi, MG dropout and others, which are not mentioned, as well as the time of computer use, but it is not mentioned in the abstract either. and I think it should be rewritten so that the result includes more information about the work done.
11. Results: the abbreviations, AVG, SLLT, iLLT, may not be so necessary but they make reading difficult. Better to just leave the abbreviation LLT, and explain average, lower and upper respectively.
12. Table one has very little information, and it is already mentioned in the text, consider eliminating it, and leaving only the text, or consider merging it with another table or including more information to make it more complete.
13. Results: the abbreviation AVG. it is confusing to consider not using it and better to describe it and better to continue using the abbreviation LLT
14. Discussion: line 181 “To analyzing the interference patterns of the tear film lipid layer can help the experts to diagnose dry eye”. Review general grammar in the manuscript. for example this statement. where should it be like this... Analyzing the interference patterns of the tear film lipid layer can help the experts to diagnose dry eye.
15. as the manuscript and the study focus on the LLT. , the degree must be focused on LLT.

---

## Round 0.2 · Minor Revisions

· Academic Editor

Minor Revisions

Final minor revisions are necessary before the manuscript is suitable for publication

·

Basic reporting

Since this is a re-review, I mention that my previous comments and questions and suggestions have been answered and clarified. With one exception:

9. Methods: write the sample size calculation for the objective of the study. Response: We have added the sample size calculation (n=55) in the background and aims in the revised MS. (Revised MS, Page 6, Line 123 ).
o Abstract: “(n=55)” the number of participants should be in result sections, not in aim , or background. Please change.

o Sample size calculation is different than the number of participants. My sugestion was to include in methos section of the manuscript a sentence about how you did the sample size calculation , in order to verify if 55 participants is enough number of participants to meets the minimum requirements to answer the research question. For example: “The sample size was calculated with XX software, based on a LLT difference between groups of XX± XX SD, to achieve an effect size (d) of XX, a significance level (α error prob) of XX, and a power (1- β err prob) of XX, in a two-tailed test, yielding a sample size of X participants in each group”

Experimental design

Since this is a re-review, I mention that my previous comments and questions and suggestions have been answered and clarified. With one exception:

9. Methods: write the sample size calculation for the objective of the study. Response: We have added the sample size calculation (n=55) in the background and aims in the revised MS. (Revised MS, Page 6, Line 123 ).
o Abstract: “(n=55)” the number of participants should be in result sections, not in aim , or background. Please change.

o Sample size calculation is different than the number of participants. My sugestion was to include in methos section of the manuscript a sentence about how you did the sample size calculation , in order to verify if 55 participants is enough number of participants to meets the minimum requirements to answer the research question. For example: “The sample size was calculated with XX software, based on a LLT difference between groups of XX± XX SD, to achieve an effect size (d) of XX, a significance level (α error prob) of XX, and a power (1- β err prob) of XX, in a two-tailed test, yielding a sample size of X participants in each group”

Validity of the findings

Since this is a re-review, I mention that my previous comments and questions and suggestions have been answered and clarified. With one exception:

9. Methods: write the sample size calculation for the objective of the study. Response: We have added the sample size calculation (n=55) in the background and aims in the revised MS. (Revised MS, Page 6, Line 123 ).
o Abstract: “(n=55)” the number of participants should be in result sections, not in aim , or background. Please change.

o Sample size calculation is different than the number of participants. My sugestion was to include in methos section of the manuscript a sentence about how you did the sample size calculation , in order to verify if 55 participants is enough number of participants to meets the minimum requirements to answer the research question. For example: “The sample size was calculated with XX software, based on a LLT difference between groups of XX± XX SD, to achieve an effect size (d) of XX, a significance level (α error prob) of XX, and a power (1- β err prob) of XX, in a two-tailed test, yielding a sample size of X participants in each group”

---

## Round 0.3 · Minor Revisions

· Academic Editor

Minor Revisions

I have now had the opportunity to read your revised manuscript, and your responses to the reviewers' comments. I believe that you have addressed the concerns raised. However, minor revisions are necessary before the manuscript is suitable for publication.

The OSDI questionnaire assesses individuals in binocular conditions (55 subjects). However, for the other variables, data from both eyes (110 eyes) were included, even though the two eyes of the same patient may be correlated. This is a common error in ophthalmology research, as the variance between eyes is generally smaller than the variance between different individuals. Consequently, combining measurements from both eyes can lead to an underestimation of the true variance, increasing the risk of a Type 1 error (see Armstrong RA, 2013). This limitation should be acknowledged in the manuscript.

Additionally, the authors mention that the MS participants worked for more than 4 hours. Please specify the type of work they performed (e.g., computer work, visually demanding tasks, patient interactions) and provide the average time spent and its standard deviation.

Finally, the manuscript does not clearly indicate whether the correlation analysis of dry eye parameters considered the ocular surface characteristics before or after work.

---

## Round 0.4 · accepted · Accept

· Academic Editor

Accept

I believe that you have addressed the concerns raised, and I am happy to accept your manuscript.